# Temporal Variations in Photosynthesis and Leaf Element Contents of ‘*Marselan*’ Grapevines in Response to Foliar Fertilizer Application

**DOI:** 10.3390/plants14060946

**Published:** 2025-03-17

**Authors:** Hai-Ju Zheng, Xin Wang, Wei-Feng Ma, Hui-Min Gou, Guo-Ping Liang, Juan Mao

**Affiliations:** The College of Horticulture, Gansu Agricultural University, Lanzhou 730070, China; 18793571207@163.com (H.-J.Z.); WX1771026708@163.com (X.W.); 18409490212@163.com (W.-F.M.); ghm1648885861@163.com (H.-M.G.); lianggp@gsau.edu.cn (G.-P.L.)

**Keywords:** foliar fertilizer, grape, photosynthetic properties, mineral elements

## Abstract

The objective of this study was to examine the impact of various foliar fertilization treatments on the growth of new shoots, photosynthetic characteristics of leaves, and mineral nutrient content in the leaves of ‘*Marselan*’ grapevines. Five distinct combinations of nano zero-valent iron (n ZVI), compound sodium nitrophenolate (CSN), and potassium dihydrogen phosphate (KH_2_PO_4_) were administered through foliar application to ‘*Marselan*’ grapevines cultivated in the Wuwei region of the Hexi Corridor, with water spray serving as the control treatment. The results showed that T5 treatment (15 mg·L^−1^ n ZVI + 0.4 g·L^−1^ CSN + 2.5 g·L^−1^ KH_2_PO_4_) significantly increased the leaf area and SPAD value of ‘*Marselan*’ grapes; T4 treatment (15 mg·L^−1^ n ZVI + 0.4 g·L^−1^ CSN + 1.67 g·L^−1^ KH_2_PO_4_) significantly increased the internode length of new grape shoots. T5 treatment was favorable to increase the basic coarseness of new grape shoots, the net photosynthetic rate of the leaves, and stomatal conductance; leaf transpiration rate was the highest under the T4 and T5 treatments; T3 (15 mg·L^−1^ n ZVI + 0.4 g·L^−1^ CSN + 1.25 g·L^−1^ KH_2_PO_4_), T4, and T5 treatments could improve leaf initial fluorescence at different periods. At 45 days after flowering, the maximum photochemical efficiency under the T3 and T4 treatments reached the highest value throughout the period, and the T3 treatment improved leaf potential maximum quantum yield. Meanwhile, the leaf nitrogen and phosphorus content under the T5 treatment were the highest in the five periods. Additionally, the contents of potassium (K), manganese (Mn), copper (Cu), and zinc (Zn) in the leaves increased significantly under the T4 and T5 treatments. The following conclusions emerged from a comprehensive analysis: the T4 treatment was the best, and the T5 treatment was the second most effective.

## 1. Introduction

Grape (*Vitis vinifera* L.), belonging to the genus Vitis of the Vitaceae family, is one of the four major fruits. It thrives in warm and sunlit environments [1,2]. The Hexi Corridor is located at a latitude of 36°~40° north and belongs to the temperate arid desert climate, with sufficient sunlight, low annual precipitation, and a large temperature difference between day and night, which is one of the wine grape growing areas in China [3,4,5]. ‘*Marselan*’ grapes exhibit robust adaptability and superior fruit quality, and are widely planted in the Wuwei area. Therefore, ‘*Marselan*’ was selected as the research object in this experiment.

However, the low soil fertility of this area, combined with the deficiency of trace elements such as iron, leads to yellowing of the plant leaves. This reduces the photosynthetic efficiency of the plants, negatively impacting the growth of wine grapes. Unreasonable soil fertilization easily causes adverse effects such as soil acidification and caking; compared with traditional fertilization methods, foliar fertilizer has the characteristics of faster absorption, more obvious effects, is less influenced by weather conditions, and is both a labor-saving and yield-boosting fertilization method [6,7]. In arid regions characterized by limited water availability, where soil fertilization necessitates adequate moisture for nutrient transport and uptake, foliar fertilization offers a rapid and direct method of delivering nutrients to various plant tissues through leaf absorption. This approach enables the precise regulation of nutrient supply tailored to the specific growth stages of plants, optimizing nutrient utilization efficiency.

Iron is an essential micronutrient for grape growth, and participates in important processes such as chlorophyll synthesis and respiration. Iron deficiency can cause symptoms such as yellow leaf disease, slow shoot development, and a reduced fruit set [8,9,10,11]. Research on the regulation of plant growth by nanomaterials has become a development trend in sustainable agriculture. Nanoscale zero-valent iron (n ZVI) is a kind of accelerator and reducing agent with superior adsorption properties and high reducibility. Previous studies have shown that 200 mg·L^−1^ n ZVI can increase the dry matter accumulation of Leonurus heterophyllus plants, and 250 mg·L^−1^ n ZVI can enhance the chlorophyll a content of Leonurus heterophyllus leaves [12]. According to Hakwon Yoon’s research, a suitable concentration of n ZVI can enhance the photosynthesis of *Arabidopsis thaliana* grown in soil, thereby increasing plant biomass [13]. Compound sodium nitrophenolate (CSN) is a common plant growth regulator. Jiang Yueshan’s research demonstrated that spraying cucumber seedlings with sodium nitrophenolate improved their growth and enhanced their cold tolerance under cold stress conditions [14]. Batool Zarina believes that spraying CSN on maize leaves under water-deficit conditions can increase the relative water content, chlorophyll content, and carotenoid content of the maize leaves, while also enhancing the yield of maize [15]. Potassium dihydrogen phosphate (KH_2_PO_4_) is a high-efficiency phosphorus potassium compound fertilizer, which is generally used to improve the transport of photosynthetic products. Experiments have been conducted using “*Cabernet Sauvignon*” grapes as experimental material. The results indicated that spraying KH_2_PO_4_ on the leaves during the color-changing period could enhance the anthocyanin content in the fruit [16]. In a study of *Stevia rebaudiana*, the author posits that low concentrations of KH_2_PO_4_ can foster plant growth, enhance the content of chlorophyll and carotenoid, and augment the photochemical properties of plants [17].

However, there are few studies which have examined the effects of different foliar fertilizer combinations on the growth of new grape shoots, leaf photosynthetic characteristics, and mineral element contents, and the relevant reports are unclear. Therefore, based on previous research, this study examines the combined application of n ZVI, CSN, and KH_2_PO_4_ from the perspectives of nutrient uptake efficiency, fertilizer efficiency, and the economic cost of foliar fertilizers. It aims to explore the effects of foliar spraying with n ZVI, CSN, and varying concentrations of KH_2_PO_4_ on shoot growth, leaf photosynthesis, and mineral element content of ‘*Marselan*’ grapes grown in the Wuwei region of the Hexi Corridor. The ultimate goal is to provide theoretical support for the rational application of different types of foliar fertilizers and to contribute to the development of the wine grape industry in this region.

## 2. Results

### 2.1. Effects of Different Treatments on Leaf Area, Leaf SPAD Value, New Shoot Base Thickness, and Internode Length of ‘Marselan’ Grape

The leaf area of grapes under different treatments at different periods was higher than that of CK; the best results were observed in the T5 treatment (Figure 1). At 45 days after flowering, the leaf area under T4 and T5 treatments reached 195.91 cm^2^ and 206.10 cm^2^, respectively, which increased by 11.31% and 17.10% compared with CK. At 60 days after flowering, the T5 treatment was significantly higher than CK, having increased by 17.10% compared with the CK treatment; 75 days after flowering, the T2, T4, and T5 treatments were significantly higher than CK, having increased by 4.23%, 10.46%, and 13.94%, respectively; at 90 days after flowering, the T4 and T5 treatments had significant differences compared with CK, having increased by 10.26% and 14.17%, respectively. To sum up, the T5 treatment had a greater impact on grape leaf area over the four stages.

Table 1 shows that the SPAD value of grape leaves under different treatments showed an overall trend of increasing first and then decreasing, reaching a peak at 90 days after flowering and then decreasing. At 45 days after flowering, the SPAD value of the T5 treatment reached the maximum, which was 43.74; at 60 days after flowering, the SPAD value of the T4 treatment was the largest, which was 46.12, followed by that of the T5 and T3 treatment, which was 45.66. At 75 days after flowering, the T4 and T5 treatments were significantly different from CK, which increased by 9.24% and 10.32%, respectively. At 90 days after flowering, the SPAD value of the T5 treatment reached the maximum of 49.02, which was 10.36% higher than that of CK. At 105 days after flowering, the SPAD value of grape leaves began to decline, but the T5 treatment was still significantly higher than CK. Therefore, spraying 15 mg·L^−1^ n ZVI + 0.4 g·L^−1^ CSN +2.5 g·L^−1^ KH_2_PO_4_(T5) on the leaves of ‘*Marselan*’ can significantly improve its SPAD value.

As shown in Table 2, at 45 days after flowering, under the T5 treatment, the internode length reached the highest value, 43.71 mm, which increased by 15.17% compared with CK. At 60 days after flowering, the T4 and T5 treatments were significantly different from CK, increasing by 20.19% and 18.14% compared with CK. From 75 to 105 days after flowering, T3, T4, and T5 were significantly different from CK. At 75 days after flowering, the maximum internode length of new shoot treated by T4 was 61.55mm, and 90 and 105 days after flowering, the maximum length of internode treated with T5 was the highest. These were 63.22 mm and 65.28 mm, respectively, increasing by 17.91% and 19.36% compared with CK. In conclusion, at 75 days after flowering, increased application of KH_2_PO_4_ was beneficial for the internode length of grape shoots, and the best concentration was 2.50 g·L^−1^.

The results in Table 3 show that there were no significant differences between the treatments at 45 to 75 days. From 90 to 105 days, T5 treatment was significantly different from CK (*p* < 0.05), and compared with CK, it increased by 18.46% and 18.94%, respectively. The result showed that spraying foliar fertilizer with 2.50g·L^−1^ KH_2_PO_4_ at 75 days after flowering was more effective in improving the base thickness of grape shoots.

### 2.2. Effects of Different Treatments on Photosynthetic Characteristics of ‘Marselan’ Grape Leaves

As shown in Figure 2A, all treatments could increase the net photosynthetic rate of grape leaves to varying degrees. From 45 d to 75 d after flowering, the T5 treatment was the highest. At 90 days after flowering, the net photosynthetic rate of leaves was higher than for the CK treatment, in the order of T4, T5, T2, T3, and T1, which increased by 56.97%, 45.95%, 40.58%, and 27.89% compared with CK, respectively. At 105 d after flowering, the highest value for the T4 treatment was 3.84 μmol·m^−2^ s^−1^.

At 45 days after flowering, the transpiration rate of the T4 and T5 treatments was significantly different from that of CK, and both were 28.89% higher than that of CK. At 60 days after flowering, the transpiration rate of each treatment was higher than that of CK. There were significant differences between T4 and T5 treatments and CK, which increased by 42.12% and 42.21%, respectively. At 75 days after flowering, the transpiration rates of T1 and T2 treatments were lower than those of CK, but there was no significant difference from CK. There was a significant difference between T5 and CK, which increased by 35.81% compared with CK. From 90 to 105 d, all treatments were higher than CK, but there was no significant difference (Figure 2B).

At 45 days after flowering, the stomatal conductance of all treatments was greater than that of CK, and T5 had the highest value. At 60 days after flowering, the value of the T5 treatment was the largest, reaching 91.23 mmol·m^−2^ s^−1^; 75 days after flowering, T5 increased by 31.27% compared with CK. The T4 treatment was second only to the T5 treatment, and it increased by 28.08% compared with CK. At 90 days after flowering, the T4 treatment was significantly different from CK, and increased by 38.50% compared with CK. At 105 days after flowering, the stomatal conductance of all treatments was as follows: T4 (53.24), T3 (52.08), T5 (47.73), CK (44.43), T2 (44.18), and T1(43.63), and there was no significant difference among all treatments (Figure 2C).

The intercellular CO_2_ concentration of grape leaves first decreased and then increased with the growth and development of the grapes (Figure 2D). At 45, 60, 90, and 105 days after flowering, there was no significant difference between the intercellular carbon dioxide value and CK. At 75 d after flowering, all treatments values were lower than those of CK, and the order from small to large was CK, T2, T1, T3, T4, and T5.

### 2.3. Effects of Different Treatments on Fluorescence Parameters of ‘Marselan’ Grape Leaves

Except for CK, the *F*_0_ value of each treatment showed a tendency of decreasing and then increasing. At 45 d after flowering, the *F*_0_ values of each treatment were, in order, T5 (469.33), T1 (334.00), T2 (290.67), CK (256.00), T3 (230.00), and T4 (223.14). At 60 days after flowering, the *F*_0_ values of all treatments except CK decreased, but there was no significant difference. At 75 d after flowering, the *F*_0_ values of all treatments were lower than CK, and T3 treatment was the lowest. At 90 days after flowering, the *F*_0_ value of T3 and T5 treatments reached their highest point—both at 242.00—which was significantly different from CK. At 105 days after flowering, the *F*_0_ values of the T3 treatment were significantly different from that of CK, and decreased by 44.53% compared with CK (Figure 3A).

At 45 days after flowering, T3 and T4 increased by 34.87% and 31.20%, respectively, compared with the CK treatment. After 60 days of flowering, the *F_m_* values of leaves were T2 (1777.33), T3 (1716.00), T5 (1607.24), CK (1490.00), T4 (1330.31), and T1 (1326.33). At 75 d after flowering, *F_m_* values of all treatments were lower than CK (799.67), and T1 (536) and T3 (540.33) were most significantly reduced. At 90 d after flowering, the T4 treatment showed a significant difference compared with CK, and compared with CK, the T4 treatment increased by 31.47%. There were no significant differences in the leaf Fm values among the treatments at 105 days after flowering (Figure 3B).

As can be seen from Figure 3C, the *F_v_*/*F_m_* value of each treated leaf showed a trend of first decreasing, then increasing and then decreasing with the growth and development of grapes. At 45 d after flowering, except for the T5 treatment, the maximum photochemical efficiency of other treatments was higher than that of CK, and the values of the T3 and T4 treatments were the largest. At 60 d after flowering, the T2 treatment showed significant improvement compared with the CK treatment. At 75 days after flowering, there was no significant difference between the treatments. At 90 days after flowering, T5 decreased by 21.74% compared with CK. At 105 days after flowering, the T3 treatment was significantly different from CK, and increased by 25.8% compared with CK.

As shown in Figure 3D, the potential maximum quantum yield of the T3 and T4 treatments was significantly higher than that of CK at 45 d after flowering—46.47% and 45.03% higher, respectively—while that of the T5 treatment was significantly lower than that of CK, decreasing by 52.36%. At 60 d after flowering, the potential maximum quantum yield of all treatments was higher than CK, and there was a significant difference between the T2 treatment and CK. At 75 d after flowering, the potential maximum quantum yield of all treatments were ordered as follows: T3, T5, T1, T2, T4, and CK. At 105 days after flowering, the T3 treatment was significantly different from CK, with an increase of 54.08% compared with CK. 

### 2.4. Effects of Different Treatments on Mineral Element Content in Grape Leaves of ‘Marselan’

As can be seen from Figure 4A, spraying different leaf fertilizers can increase the nitrogen content in grape leaves to different degrees. At 45 days after flowering, the nitrogen content of leaves was in the order of T5, T2, T4, T1, T3, and CK. Compared with CK, the nitrogen content of T5 and T2 was significantly increased, by 29.14% and 25.88%, respectively. At 60 days after flowering, the nitrogen content of leaves treated with T5 was the highest, reaching 21.3 g·kg^−1^. From 75 to 105 days after flowering, the nitrogen content of leaves under the T5 treatment was the highest, which increased by 29.46%, 32.19%, and 30.88% compared with CK, respectively.

The phosphorus content in leaves showed a slow declining trend with the growth and development of grapes (Figure 4B). At 45 days after flowering, the phosphorus content in leaves of all treatments was lower than CK except for the T5 treatment, but the difference was not significant. From 60 to 90 days after flowering, the leaf phosphorus content was the highest under the T5 treatment. At 105 days after flowering, the phosphorus content of CK and the T1 treatment was essentially the same, at 2.81 and 2.79 g·kg^−1^, respectively. The content of the T3 treatment was the lowest, and that of the T5 treatment was the highest, which increased by 24.05% compared with CK.

The potassium content in leaves treated with T3, T4, and T5 was significantly different from that of CK at 45 days after flowering, increasing by 20.13%, 21.59%, and 19.48%, respectively (Figure 4C). From 60 to 90 days after flowering, potassium content in leaves of all treatments was higher than that of CK, and there were significant differences between T4 and T5 treatments and CK. At 105 days after flowering, potassium content in leaves was lower on the whole. The potassium content of each treatment was ranked from highest to lowest as T4 (10.60 g·kg^−1^), T5 (10.55 g·kg^−1^), T3 (9.19 g·kg^−1^), T2 (8.01 g·kg^−1^), CK (7.56 g·kg^−1^), and T1 (7.17 g·kg^−1^).

As can be seen from Figure 5A at 45 days after flowering, all treatments except T1 showed significant differences from CK. At 60 days after flowering, there was no significant difference among the treatments, and the ranking from lowest to highest was T5, T4, T3, T2, T1, and CK. At 75 days after flowering, the calcium content of the T2 treatment was the lowest, which decreased by 13.94% compared with CK, and the T3 treatment was the highest, which increased by 9.47% compared with CK. From 90 to 105 days after flowering, the calcium content of leaves showed that the CK value was higher than other treatments, and there was no significant difference among the five treatments.

As can be seen from Figure 5B, magnesium content in the leaves of each treatment was relatively stable during the whole development period. At 45 days after flowering, all treatment values were higher than CK, but there was no significant difference between groups. From 60 to 75 days after flowering, the magnesium content of each treatment was essentially the same and there was no significant difference. At 90 days after flowering, the magnesium content of each treated leaf was in the order of T1, T2, CK, T3, T4, and T5. At 105 days after flowering, the magnesium content in the leaves of each fertilizer treatment was lower than that of CK, and the T2 treatment was the highest, which was 23.21% higher than CK.

We can see from Figure 5C that the iron content in grape leaves showed a gradual increasing trend. At 45 days after flowering, there was no significant difference between the treatments. At 60 days after flowering, the T2 treatment was significantly different from CK, and increased by 15.40% compared with CK. At 75 days after flowering, the treatments increased by 11.02%, 16.00%, 18.93%, 23.50%, and 19.20%, respectively, compared with CK. At 90 days after flowering, the highest iron content in T3 treated leaves was 251.69 mg·kg^−1^. At 105 days after flowering, T3 and T4 treatments were significantly different from CK, increasing by 22.52% and 16.91% compared with CK, respectively.

As can be seen from Figure 5D, compared with CK, manganese content in the leaves of each treatment increased from 45 to 60 days after flowering, but there was no significant difference among the treatments. At 75 days after flowering, the manganese content of T4 and T5 treatment had significantly increased, by 11.48% and 12.14%, respectively. At 90 days after flowering, the Mn content of each treated leaf reached the peak value of its growth period, and the Mn content of T5 treated leaves reached a highest value of 157.8 mg·kg^−1^. At 105 days after flowering, the T4 treatment was significantly different from CK, and increased by 15.84% compared with CK.

The copper content of each treated leaf showed a trend of first increasing and then decreasing (Figure 5E). At 45 days after flowering, the copper content of all treatment groups was basically the same, and there was no significant difference among groups. At 60 days after flowering, T3, T4, and T5 were significantly different compared to CK, increasing by 22.62%, 22.89%, and 24.88% compared with CK, respectively. At 75 days after flowering, the copper content of leaves treated with T1, T2, and T3 was lower than that of CK, and the copper content of leaves treated with T5 was significantly different from CK, increasing by 6.06%. At 90 days after flowering, all treatments except T1 were significantly different from CK. At 105 days after flowering, the copper content in leaves was, in order, T4 (44.90 mg·kg^−1^), T5 (44.72 mg·kg^−1^), T3 (41.58 mg·kg^−1^), T2 (37.70 mg·kg^−1^), CK (31.20 mg·kg^−1^), and T1 (30.49 mg·kg^−1^).

Zinc content in leaves increased first and then decreased with the growth and development of grapes (Figure 5F). At 45 days after flowering, there was no significant difference in zinc content among the treatments; at 60 days after flowering, the content of zinc began to show significant differences, ranging from 26.69 mg·kg^−1^ to 35.33 mg·kg^−1^; at 75 days after flowering, the zinc content of the T4 and T5 treatments was significantly different from that of CK, and the zinc content of T5 was the highest, which was 42.71 mg·kg^−1^, followed by T4, which was 39.11 mg·kg^−1^, increasing by 24.27% and 17.28% compared with CK, respectively. There was no significant difference in the content of copper in leaves of all treatments from 90 to 105 days after flowering.

### 2.5. Comprehensive Evaluation of Shoot Growth, Leaf Photosynthetic Characteristics, and Mineral Element Content of ‘Marselan’ Grape Under Different Treatments

The evaluation system was established according to the measured index values, and the relevant indices of each treatment and each period were selected for evaluation. Quantitative conversion was performed to obtain the membership function values of the data corresponding to each indicator in the interval [0, 1] (Table 4), and the comprehensive evaluation ranking was T4, T5, T3, T2, T1, and CK.

## 3. Discussion

The growth and development of plants are affected by many factors, such as water, climate, and nutrients. Ma Qinghua’s [18] research showed that the appropriate concentration of KH_2_PO_4_ solution treatment on rose soil and leaves in different flowering cycles would promote the rose diameter, chlorophyll content, and total root length. The results of this study showed that different combinations of foliar fertilizers could significantly improve the leaf area (Figure 1), chlorophyll relative content (Table 1), shoot base thickness (Table 2), and internode length (Table 3) of grape plants. The application of CSN based on n ZVI could increase leaf SPAD values and improve the transpiration efficiency at maturity. After increasing the concentration of KH_2_PO_4_, the plant growth status was further improved, and the higher concentration of KH_2_PO_4_ was better. This is because phosphorus and potassium are directly involved in chlorophyll synthesis, cell division, and cell expansion, which was consistent with previous research.

Leaf photosynthesis is not only affected by the external environment, but is also related to various mechanisms in leaves [19,20,21]. Previous studies showed that the levels of nitrogen, phosphorus, and potassium could affect the net photosynthetic rate, transpiration rate, and stomatal conductance of henna [22]. In a study of tea, it was found that foliar application of potassium dihydrogen phosphate could increase the proportion of stomatal opening in tea leaves, thus promoting the growth and development of tea [23]. In this study, the promotion effect of potassium dihydrogen phosphate (T3–T5) on net photosynthetic rate and transpiration rate increased with the increase in concentration, and there were significant changes with the phenological period. The photosynthesis performance of T5 and T4 treatments was better than that of other treatments in most cases. When the grape entered the color-changing period (90 days after flowering), its photosynthetic activity began to weaken due to the change in the plant growth cycle (Figure 2). At this stage, the stomata gradually closed, reducing the absorption of carbon dioxide.

Chlorophyll fluorescence measurement is a tool to detect the stress level of plants [24]. *F*_0_ was only related to the chlorophyll concentration of leaves, *F_m_* is used to describe the photoelectron transfer efficiency of the *PSII* system, which reflects photosynthetic characteristics. *F_v_*/*F_m_* and *F_v_*/*F*_0_ are important indicators to measure the photosynthetic performance of plants. The larger the value, the better the photosynthetic efficiency. Studies have shown that under phosphorus deficiency stress, the maximum photochemical efficiency of *PSII* in violet was significantly lower than that in *Brassica napus* [25]. Changes in initial fluorescence (*F*_0_) were observed. From 45 to 60 days after flowering, the influence trend of each treatment on *F*_0_ was the same, while from 75 to 105 days after flowering, the *F*_0_ values of the T1 treatment were significantly lower than that of the CK treatment, while other treatments had obvious fluctuations. The analysis of maximum fluorescence (*F_m_*) showed that the treatment containing potassium dihydrogen phosphate in the spraying treatment had a significant effect on the improvement of *F_m_* value at the early stage of plant growth, and the T4 treatment was significantly higher than most other treatments. Under different treatments, the maximum photochemical efficiency (*F_v_*/*F_m_*) and potential maximum quantum yield of *PSII* (*F_v_*/*F*_0_) showed varying degrees of changes during each phenological period, especially from 75 to 105 days after flowering, where the improvement of *F_v_*/*F_m_* and *F_v_*/*F*_0_ values by the T3 and T4 treatments showed the potential to promote photosynthetic efficiency (Figure 3).

Nitrogen is generally considered to be the most important factor limiting plant growth [26,27,28], is also the basic element of important substances such as protein, nucleic acid, and amino acid in plants, and has a significant role in promoting the germination of buds and the growth of branches and leaves [29,30]. This study found that foliar fertilizer spraying can effectively improve the nitrogen content of grape leaves. Among them, the T5 treatment, with a high concentration of potassium dihydrogen phosphate, was the best, while the T3 treatment, with a low concentration of potassium dihydrogen phosphate, has a certain inhibitory effect on the absorption of nitrogen in the early stage (Figure 4A), which is similar to Hu Wenjie’s conclusion that compound fertilizer potassium nitrate can improve the content of total nitrogen in grape plants to a certain extent, while potassium sulfate and potassium dihydrogen phosphate treatments reduce the content [31].

Arrobas’ research suggested that phosphorus was involved in the construction of grape morphology in the early stages of grape growth [32]. The phosphorus content in leaves showed a decreasing trend in the whole growth cycle. In this study, a high concentration of potassium dihydrogen phosphate can significantly increase the phosphorus content, and the application of sodium nitrophenolate combined with potassium dihydrogen phosphate on the basis of a single application of nano zero-valent iron can slow down the reduction rate of phosphorus (Figure 4B). Potassium is one of the three essential elements for plants. It mainly exists in the form of free K^+^ in plants. It can regulate the movement of stomata, participate in the transportation of phloem assimilates, and promote the assimilation of fruit trees [33,34,35]. At the same time, this study found that the potassium content in leaves generally increased initially and then decreased. Compared with the CK treatment, the potassium content in leaves of each treatment group was higher, and nano zero-valent iron could promote the absorption of potassium. On this basis, increasing the appropriate concentration of potassium dihydrogen phosphate was more conducive to the accumulation of potassium in leaves, and the T4 treatment had a continuous effect on promoting the absorption of potassium in leaves throughout the growth period (Figure 4C). KH_2_PO_4_ is a highly water-soluble, high-quality phosphorus and potassium fertilizer that can directly increase the content of phosphorus and potassium in plant leaves. In addition to its direct nutrient supply, potassium ions from KH_2_PO_4_ can enhance the activity of enzymes related to nitrogen metabolism, such as nitrate reductase and glutamine synthetase. This enzymatic activation promotes nitrogen uptake and assimilation in plants, thereby improving overall nutrient utilization efficiency.

Trace elements, such as calcium, magnesium, iron, copper, manganese, and zinc, they also play an indispensable role in the growth and development of grapes [36,37]. In this experiment, the calcium content of leaves in the control group increased slowly with growth, while that in the fertilization group increased first and then decreased, and changed significantly at a high concentration of fertilization (Figure 5A). Magnesium is an important component of chlorophyll and plays a key role in photosynthesis [38,39]. This study found that the magnesium content in leaves of the control group continued to rise, while that of the fertilization group increased first and then decreased (Figure 5B). The magnesium content remained essentially unchanged, likely attributable to magnesium’s relatively low mobility within plants. Magnesium is predominantly absorbed through the roots and translocated via the xylem, rendering foliar fertilization less effective in altering its foliar concentration. Zinc participates in the activation, catalysis, and stabilization of more than 300 enzymes in plants, and is an indispensable trace element for plant growth and development [40]. The results showed that foliar fertilizer could increase the iron content of grape leaves, and the T4 treatment had the best effect. Foliar fertilizer also significantly increased the copper content of leaves, and the effect was most obvious from the expansion stage to the color conversion stage. The change in manganese content was similar to that of copper, which was kept stable by foliar fertilizer, and increased significantly in the color conversion period. The zinc content in leaves of the fertilization treatments was mostly higher than that of the control group, indicating that foliar fertilizer was helpful to maintain the stability of zinc content (Figure 5C–F). Approximately 60 days after flowering, grapevines enter the fruit expansion stage, during which the leaves exhibit an increased demand for copper and zinc. These micronutrients are essential for promoting fruit development and enhancing photosynthetic efficiency. Copper plays a significant role in photosynthetic processes, including chlorophyll synthesis and electron transport, while zinc is crucial for enzyme activation and carbohydrate metabolism, both of which are vital for fruit growth and quality. The effect of foliar fertilizer formula on the content of elements in grape leaves was positive, but the degree and period of influence varied with elements. With an increase in the potassium fertilizer gradient, it will significantly affect the content of nitrogen, potassium, calcium, and magnesium in leaves. The relationship between mineral element content and fertilization rate is complex and affected by many factors [41,42,43]. A single application of nano zero-valent iron could promote the absorption of nitrogen, potassium, magnesium, and manganese by plants. Therefore, the absorption of calcium and magnesium in leaves by potassium dihydrogen phosphate first increased and then decreased with plant growth, while the absorption of iron showed a contrary trend. At 90 days after flowering, a pivotal phase in grape berry maturation, nutrient allocation is predominantly redirected towards fruit development, thereby inducing a decline in foliar calcium concentration.

## 4. Materials and Methods

### 4.1. Environmental Characteristics of the Test Area

This experiment was carried out at Yinuo Winery in Wuwei City, Gansu Province (longitude 106°26′24″ east longitude, latitude 35°12′18″ north latitude). The soil in the experimental park is neutral to weakly alkaline sandy loam. The average annual rainfall is between 150 and 247 mm, the average annual sunshine duration reaches 2876.9 h, the average annual temperature is 7.1 °C, and the average annual precipitation is about 123 mm.

### 4.2. Test Materials and Design

In this experiment, 4-year-old ‘*Marselan*’ grape were used as the test material, and a randomized block design was adopted. Through the pre-test, we found that the most suitable nano zero-valent iron concentration for wine grapes in this region is 15 mg·L^−1^. Production practice shows that the safe spraying concentration of potassium dihydrogen phosphate for berry crops is between a 400- and 800-fold dilution; compound sodium nitrophenolate is safe and harmless within the concentration range of a 1200–3000-fold dilution. There were five treatments and spraying water was used as the control (Table 5). Foliar fertilizer was sprayed from 45 days after flowering, and completed before 9:00 a.m on sunny days. Three replications were made for each treatment with 16 plants per replication, and each replication was sprayed with 10 L per application.

### 4.3. Determination of Indexes and Methods

#### 4.3.1. Determination of Root Thickness, Internode Length, Leaf SPAD Value, and Leaf Area of Grape

Vernier calipers were used to measure the thickness of the base of new shoots, the length of the fifth internode of branches was measured with a tape measure, and 30 leaves (the 5th leaf from the base upwards) were measured for each randomly selected part; the leaf area was measured with TPYX-A (Zhejiang, China), a crop leaf morphology tester. The SPAD value of the leaves was determined with the handheld Topper TYS-B portable SPAD analyzer (accuracy ± 1.0, Zhejiang, China).

#### 4.3.2. Determination of Photosynthetic Parameters in Grape Leaves

Leaf photosynthetic parameters: net photosynthetic rate (*P_n_*), stomatal conductance (*G_s_*), transpiration rate (*T_r_*), and intercellular CO_2_ concentration (*C_i_*) were determined under natural light using the LI-6400 portable photosynthetic apparatus (Beijing, China).

#### 4.3.3. Determination of Fluorescence Parameters of Grape Leaves

The grape leaves (the fifth leaf from the base up) were fully dark-adapted for 30 min; a Junier-PAM chlorophyll fluorescence analyzer (Walz, Germany) was used to determine the fluorescence parameters of the leaves: *F*_0_ (basal fluorescence), *F_v_* (variable fluorescence), *F_v_/F_m_* (maximum photochemical quantum yield), and *F_v_/F*_0_ (potential maximum quantum yield).

#### 4.3.4. Determination of Element Content in Leaves

At 45, 60, 75, 90, and 105 days after flowering, 30 leaves’ adjacent functional leaves were randomly collected for each treatment, respectively. After the collected leaves were killed, dried, ground, and sieved, 0.5 g was accurately weighed for digestion. After digestion, the total nitrogen content in the sample was determined by Kjeldahl nitrogen analyzer (Shandong, China). The content of total phosphorus was determined by molybdenum antimony anti-colorimetry. The contents of calcium, copper, iron, magnesium, manganese, potassium, and zinc were determined by atomic absorption spectrometry [44].

### 4.4. Data Processing

Excel 2021 was used for data statistics; SPSS 22.0 was used for correlation analysis and one-way analysis of variance to compare the difference between different treatments. The significance analysis was conducted using Duncan’s multiple range test. (*p* < 0.05). Graphs were drawn using Origin 2022 software.

## 5. Conclusions

As shown in Figure 6, spraying foliar fertilizers with different formulas can improve the leaf area of grapes’ SPAD, leaf fluorescence parameters, basal thickness of the new shoots, and internode length of the new shoots of grapes, and promote the increase in leaf photosynthetic rate and transpiration rate, as well as significantly increase the content of mineral elements in the leaves. Comprehensive analysis showed that T4 (15 mg·L^−1^ n ZVI + 0.4 g·L^−1^ CSN + 1.67 g·L^−1^ KH_2_PO_4_) was the best, followed by T5 (15 mg·L^−1^ n ZVI + 0.4 g·L^−1^ CSN 2.5 g·L^−1^ KH_2_PO_4_).

## Figures and Tables

**Figure 1 plants-14-00946-f001:**
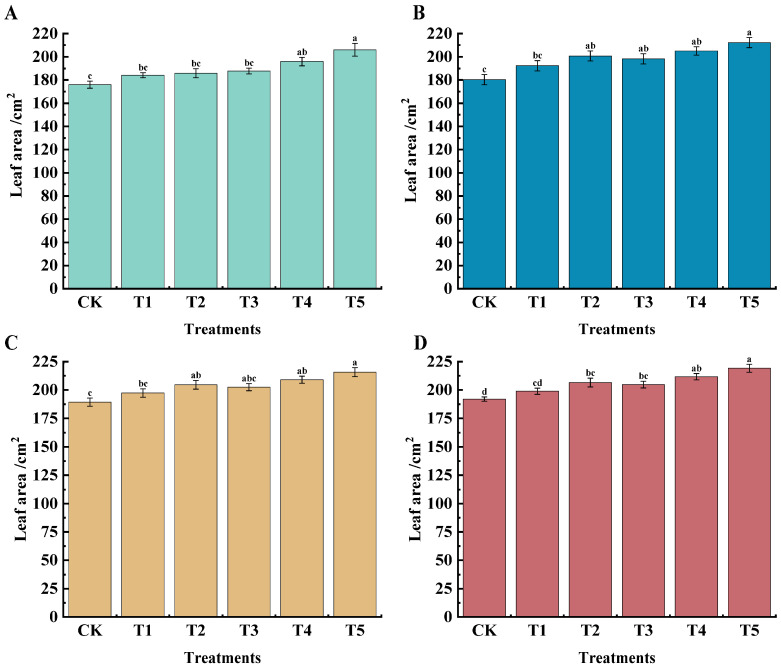
Effects of different treatments on grape leaf area of ‘*Marselan*’ grapes. Unit: cm^2^. (**A**) indicates the leaf area of each treatment at 45 d after flowering, (**B**) indicates the leaf area of each treatment at 60 d after flowering, (**C**) indicates the leaf area of each treatment at 75 d after flowering, and (**D**) indicates the leaf area of each treatment at 90 d after flowering. Statistical significance was analyzed via one-way ANOVA. Error bars represent the mean ± SE from three biological repeats. Different lowercase letters denote significant differences, whereas the same lowercase letters indicate no statistical difference (*p* < 0.05). CK indicates water spray, T1 indicates (15 mg·L^−1^ n ZVI), T2 indicates (15 mg·L^−1^ n ZVI + 0.4 g·L^−1^ CSN), T3 indicates (15 mg·L^−1^n ZVI + 0.4 g·L^−1^ CSN + 1.25 g·L^−1^ KH_2_PO_4_), T4 indicates (15 mg·L^−1^ n ZVI + 0.4 g·L^−1^ CSN + 1.67 g·L^−1^ KH_2_PO_4_), T5 indicates (15 mg·L^−1^ n ZVI + 0.4 g·L^−1^ CSN + 2.5 g·L^−1^ KH_2_PO_4_).

**Figure 2 plants-14-00946-f002:**
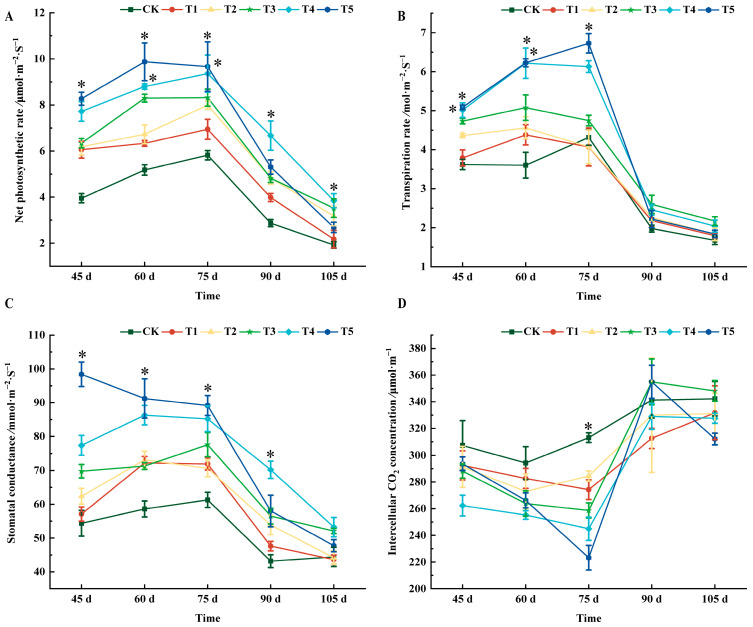
Effects of different treatments on photosynthetic characteristics of ‘*Marselan*’ grape leaves. (**A**) indicates the net photosynthetic rate of leaves in five periods, (**B**) indicates the transpiration rate of leaves in five periods, (**C**) indicates the stomatal conductance of leaves in four periods, and (**D**) indicates the intercellular CO_2_ concentration of leaves in five periods. The ‘*’ sign indicates that there is saliency between each treatment and CK at *p* < 0.05. Error bars represent the mean ± SE from three biological repeats.

**Figure 3 plants-14-00946-f003:**
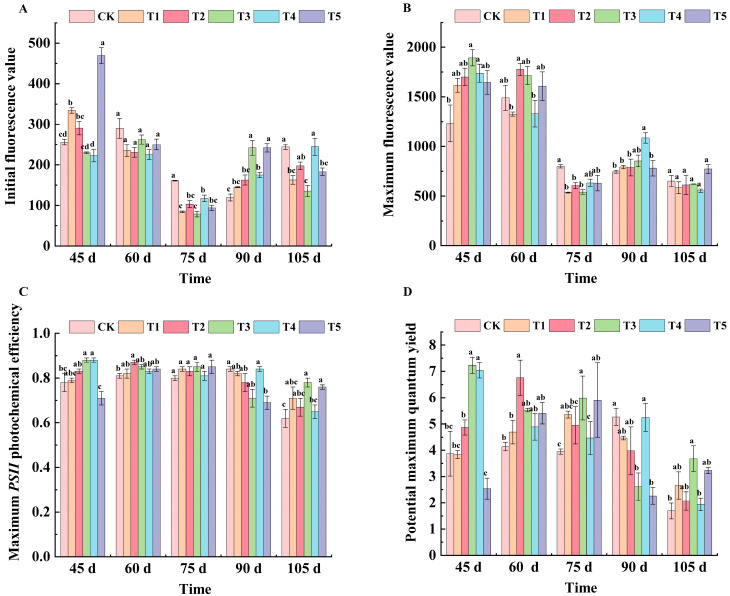
Effects of different treatments on fluorescence parameters of ‘*Marselan*’ grape leaves. In the figure, (**A**) indicates the initial fluorescence value of leaves in five periods, (**B**) indicates the maximum fluorescence value of leaves in five periods, (**C**) indicates the maximum *PSII* photochemical efficiency of leaves in five periods, and (**D**) indicates the potential maximum quantum yield of leaves in five periods. Error bars represent the mean ± SE from three biological repeats. Different lowercase letters denote significant differences, whereas the same lowercase letters indicate no statistical difference.

**Figure 4 plants-14-00946-f004:**
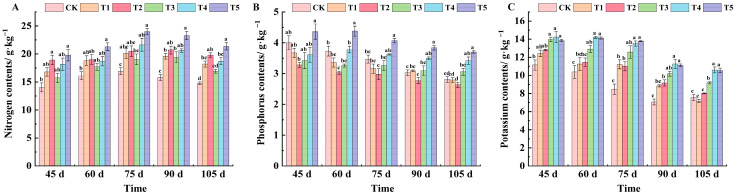
Effects of different treatments on the contents of nitrogen, phosphorus, and potassium in the leaves of ‘*Marselan*’ grape. (**A**) represents the amount of nitrogen in leaves in 5 periods, (**B**) represents the amount of phosphorus in leaves in 5 periods, and (**C**) represents the amount of potassium in leaves in 5 periods. Error bars represent the mean ± SE from three biological repeats. Different lowercase letters denote significant differences, whereas the same lowercase letters indicate no statistical difference.

**Figure 5 plants-14-00946-f005:**
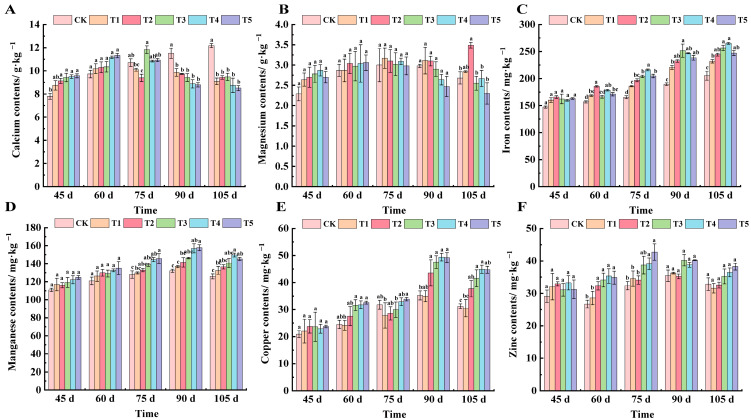
Effect of different treatments on the content of calcium, magnesium, iron, manganese, copper, and zinc in the leaves of ‘*Marselan*’ grape. (**A**) represents the calcium content of leaves in 5 periods, (**B**) represents the magnesium content of leaves in 5 periods, (**C**) represents the iron content of leaves in 5 periods, (**D**) represents the manganese content of leaves in 5 periods, (**E**) represents the copper content of leaves in 5 periods, and (**F**) represents the zinc content of leaves in 5 periods. Error bars represent the mean ± SE from three biological repeats. Different lowercase letters denote significant differences, whereas the same lowercase letters indicate no statistical difference.

**Figure 6 plants-14-00946-f006:**
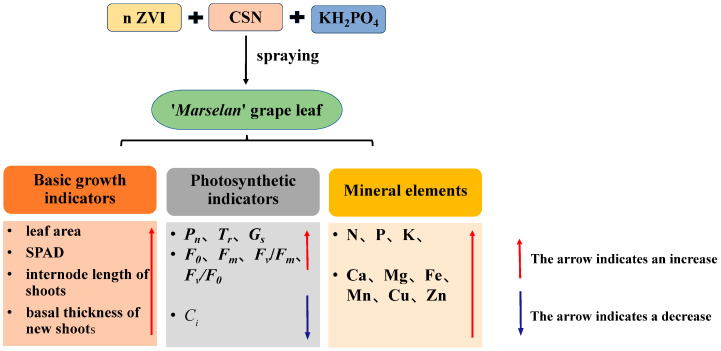
The effect of different foliar fertilizers on the growth of ‘*Marselan*’ grapevines.

**Table 1 plants-14-00946-t001:** Effects of different treatments on SPAD values of ‘*Marselan*’ grape leaves.

Treatments	45 d	60 d	75 d	90 d	105 d
CK	38.46 ± 0.65 c	41.92 ± 0.52 b	43.82 ± 0.92 c	43.94 ± 0.44 c	40.42 ± 0.59 d
T1	41.52 ± 0.29 b	44.48 ± 0.68 a	46.22 ± 0.80 ab	46.54 ± 0.56 b	41.20 ± 0.29 cd
T2	42.04 ± 0.506 ab	44.22 ± 0.66 a	46.32 ± 0.74 bc	46.98 ± 0.43 ab	43.32 ± 0.63 ab
T3	42.64 ± 0.68 ab	45.66 ± 0.60 a	46.12 ± 0.95 ab	47.36 ± 0.83 ab	43.00 ± 0.99 bc
T4	42.52 ± 0.73 a	46.12 ± 0.95 a	48.28 ± 0.72 a	48.66 ± 0.67 a	42.92 ± 0.40 abc
T5	43.74 ± 0.79 a	45.66 ± 0.60 a	48.86 ± 0.60 a	49.02 ± 0.84 a	45.10 ± 0.58 a

Statistical significance was analyzed via one-way ANOVA. Values in tables represent the mean ± SE from three biological repeats. Different lowercase letters denote significant differences, whereas the same lowercase letters indicate no statistical difference (*p* < 0.05). CK indicates water spray, T1 indicates (15 mg·L^−1^ n ZVI), T2 indicates (15 mg·L^−1^ n ZVI + 0.4 g·L^−1^ CSN), T3 indicates (15 mg·L^−1^n ZVI + 0.4 g·L^−1^ CSN + 1.25 g·L^−1^ KH_2_PO_4_), T4 indicates (15 mg·L^−1^ n ZVI + 0.4 g·L^−1^ CSN + 1.67 g·L^−1^ KH_2_PO_4_), T5 indicates (15 mg·L^−1^ n ZVI + 0.4 g·L^−1^ CSN + 2.5 g·L^−1^ KH_2_PO_4_).

**Table 2 plants-14-00946-t002:** Effects of different treatments on internode length of ‘*Marselan*’ grape shoots; unit: mm.

Treatments	45 d	60 d	75 d	90 d	105 d
CK	37.08 ± 2.60 a	46.16 ± 2.13 c	50.02 ± 1.85 c	51.90 ± 1.75 b	52.64 ± 1.67 c
T1	39.20 ± 1.99 a	49.59 ± 2.32 bc	54.80 ± 2.30 bc	56.50 ± 2.43 b	57.75 ± 2.35 bc
T2	41.64 ± 0.70 a	51.53 ± 1.02 abc	54.17 ± 1.34 bc	55.46 ± 1.53 b	56.72 ± 1.53 c
T3	43.05 ± 0.91 a	55.20 ± 1.13 ab	61.34 ± 1.01 a	62.69 ± 1.23 a	63.63 ± 1.39 a
T4	42.49 ± 1.97 a	57.84 ± 0.60 a	61.55 ± 0.80 a	63.08 ± 0.79 a	64.72 ± 0.58 a
T5	43.71 ± 1.54 a	56.39 ± 0.81 a	60.56 ± 1.02 a	63.22 ± 0.91 a	65.28 ± 0.87 a

Statistical significance was analyzed via one-way ANOVA. Values in tables represent the mean ± SE from three biological repeats. Different lowercase letters denote significant differences, whereas the same lowercase letters indicate no statistical difference (*p* < 0.05). CK indicates water spray, T1 indicates (15 mg·L^−1^ n ZVI), T2 indicates (15 mg·L^−1^ n ZVI + 0.4 g·L^−1^ CSN), T3 indicates (15 mg·L^−1^n ZVI + 0.4 g·L^−1^ CSN + 1.25 g·L^−1^ KH_2_PO_4_), T4 indicates (15 mg·L^−1^ n ZVI + 0.4 g·L^−1^ CSN + 1.67 g·L^−1^ KH_2_PO_4_), T5 indicates (15 mg·L^−1^ n ZVI + 0.4 g·L^−1^ CSN + 2.5 g·L^−1^ KH_2_PO_4_).

**Table 3 plants-14-00946-t003:** Effect of different treatments on basal thickness of new shoots of ‘*Marselan*’ grape; unit: mm.

Treatments	45 d	60 d	75 d	90 d	105 d
CK	8.75 ± 0.69 a	10.20 ± 0.56 a	10.84 ± 0.58 a	11.35 ± 0.62 b	11.77 ± 0.63 c
T1	9.14 ± 0.67 a	10.17 ± 0.68 a	10.88 ± 0.63 a	11.52 ± 0.69 b	12.02 ± 0.65 bc
T2	9.42 ± 0.33 a	10.50 ± 0.33 a	11.33 ± 0.39 a	11.83 ± 0.38 ab	12.21 ± 0.38 bc
T3	9.47 ± 0.63 a	10.46 ± 0.47 a	12.16 ± 0.35 a	12.91 ± 0.29 ab	13.34 ± 0.27 abc
T4	9.86 ± 0.43 a	11.41 ± 0.47 a	12.94 ± 0.47 a	13.62 ± 0.34 ab	13.91 ± 0.28 ab
T5	10.29 ± 0.94 a	11.81 ± 0.77 a	12.66 ± 0.54 a	13.92 ± 0.71 a	14.52 ± 0.55 a

Statistical significance was analyzed via one-way ANOVA. Values in tables represent the mean ± SE from three biological repeats. Different lowercase letters denote significant differences, whereas the same lowercase letters indicate no statistical difference (*p* < 0.05). CK indicates water pray, T1 indicates (15 mg·L^−1^ n ZVI), T2 indicates (15 mg·L^−1^ n ZVI + 0.4 g·L^−1^ CSN), T3 indicates (15 mg·L^−1^n ZVI + 0.4 g·L^−1^ CSN + 1.25 g·L^−1^ KH_2_PO_4_), T4 indicates (15 mg·L^−1^ n ZVI + 0.4 g·L^−1^ CSN + 1.67 g·L^−1^ KH_2_PO_4_), T5 indicates (15 mg·L^−1^ n ZVI + 0.4 g·L^−1^ CSN + 2.5 g·L^−1^ KH_2_PO_4_).

**Table 4 plants-14-00946-t004:** Comprehensive evaluation of different treatments on the growth of new shoots, photosynthetic characteristics of leaves, and their content of mineral elements in ‘*Marselan*’ grapes.

Treatments	CK	T1	T2	T3	T4	T5
Leaf area	0.00	0.28	0.53	0.48	0.73	1.00
SPAD	0.00	0.39	0.61	0.64	0.77	1.00
Internode length of new shoots	0.00	0.40	0.35	0.92	0.98	1.00
Roughness at the base of new shoots	0.00	0.08	0.19	0.57	0.84	1.00
Rate of transpiration	0.00	0.24	0.37	0.89	1.00	0.82
Intercellular CO_2_ concentration	0.96	0.04	0.35	1.00	0.00	0.25
Net photosynthetic rate	0.00	0.28	0.56	0.65	1.00	0.71
Stomatal conductance	0.00	0.17	0.33	0.59	1.00	0.74
Initial fluorescence value	0.45	0.00	0.33	0.39	0.68	1.00
Maximum fluorescence value	0.09	0.00	0.40	0.68	1.00	0.79
Maximum photochemical efficiency of *PSII*	0.19	1.00	0.32	0.00	0.71	0.10
Potential maximum quantum yield	0.54	0.77	0.48	0.76	1.00	0.00
Nitrogen content	0.00	0.51	0.69	0.39	0.62	1.00
Phosphorus content	0.28	0.23	0.00	0.31	0.66	1.00
Potassium content	0.00	0.24	0.37	0.69	1.00	0.98
Calcium content	0.16	0.19	0.21	1.00	0.04	0.00
Magnesium content	0.43	0.64	1.00	0.37	0.32	0.00
Iron content	0.00	0.50	0.72	0.94	1.00	0.78
Manganese content	0.00	0.24	0.42	0.61	1.00	0.95
Copper content	0.05	0.00	0.54	0.82	1.00	1.00
Zinc content	0.00	0.05	0.10	0.72	0.75	1.00
Global affiliation function	0.15	0.30	0.42	0.64	0.77	0.72
Comprehensive ranking	6	5	4	3	1	2

**Table 5 plants-14-00946-t005:** Foliar fertilization spray program.

Treatments	Nano Zero-Valent Iron (mg·L^−1^)	Compound Sodium Nitrophenolate (g·L^−1^)	Potassium Dihydrogen Phosphate (g·L^−1^)
T1 (Treatment 1)	15	0	0
T2 (Treatment 2)	15	0.4	0
T3 (Treatment 3)	15	0.4	1.25
T4 (Treatment 4)	15	0.4	1.67
T5 (Treatment 5)	15	0.4	2.50
CK (Control)	0	0	0

## Data Availability

Data are contained within the article.

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
