# Peer review of "Temporal Variations in Photosynthesis and Leaf Element Contents of ‘*Marselan*’ Grapevines in Response to Foliar Fertilizer Application"

_plants, 2025, doi:10.3390/plants14060946_

Round 1

Reviewer 1 Report

Comments and Suggestions for Authors

The authors prepared an interesting manuscript titled Temporal Variations in Photosynthesis and Leaf Element Contents of ‘Marselan’ Grapevines in Response to Foliar Fertilizer Application. This study investigated the effects of five distinct foliar fertilization treatments. 

Some specific comments:

Abstract. The first sentence is too long. Make it into a few sentences, one with a highlighted aim of the study.   Introduction. Check the citations. It is wrong. Why is important foliar fertilization? What about other types of fertilization?   Results. Uniform the text size in figures. What shoes error bars in figures? More information is needed about the figures ( like under Table 1). Tables 2 and 3 are missing information about statistics. It is only under Table 1.   Figure 4. Too confusing and unclear. Rethink the presentation of data. Currently, many graphs are difficult to understand and there is a lot of confusion about what is shown. Figure 5 I don't understand this figure... You talk about significant differences but I can't see any statistical data. The size of the part are showing something?   Discussion. Also check the citations. And when you talk about your research data, indicate in which table or figure they are presented.   Materials and Methods. Clearly explain the treatments and abbreviations, lines 418 - 420.   Figure 6. Must be "The effect..."    

Comments on the Quality of English Language

There are numerous spelling mistakes and issues with sentence structure. This text needs to be reviewed and corrected.

Reviewer 2 Report

Comments and Suggestions for Authors

Dear Authors,

After reviewing the manuscript, my recommendations are:

The abstract identifies the purpose of the study, namely the investigation of the effect of foliar fertilization treatments on the mineral content of plants and grape berries, as well as on photosynthetic characteristics. The main obtained results and the final conclusion of the study are mentioned. However there are areas for improvement.

Lines 21-26: split the phrase into separate sentences.

The introduction provides information that contributes to the understanding of the study, but improvements are possible for greater clarity and accessibility for readers.

A brief explanation of why the Marselan variety was chosen for the study.

Why foliar fertilizer is more suitable than soil application in the arid climate from Hexi Corridor.

Please clarify if some observed differences—such as changes in phosphorus content—are statistically significant.

With a heavy emphasis on quantitative data, statistical validation, and well-structured tables and figures, the results are presented in an understandable manner and formatted suitably.

Results

Paragraph. 2.1.

The two figures and three tables that are provided are useful.to help visualise the effects of different treatments on the thickness of the shoot base, the length of the internode, the area of the leaf, and the SPAD values

Please specify whether the error bars in the figure reflect standard error (SE) or standard deviation (SD).

To minimise uncertainty, indicate whether percentage changes are calculated relative to CK or the preceding measurement time point.

A few words of the agronomic implications: why does T5 perform best in most criteria, and how does this apply to real vineyard management?

Paragraph 2.2.

The text uses the term "significant difference" but does not define the statistical method (e.g., ANOVA, t-test) or the significance level (p < 0.05; p < 0.01;…). As a result, statistical significance claims become ambiguous and unreliable.

"compared to CK" and "there was a significant difference" are used often without offering fresh information.

Lines 173-175: "At 90 days after flowering, the F₀ value of T3 and T5 treatments reached the highest, both were 242.00, which was significantly different from CK." - It is scientifically uncertain why T3 and T5 have the greatest F₀ levels. Increased F₀ usually implies photoinhibition or damage to PSII. There is not an explanation for the growth.

Lines 175-176: At 105 days after flowering, the F0 values of T3 treatment was significantly dif-175 ferent from that of CK, and decreased by 80.28% compared with CK (Figure 3A). - The 80.28% drop in F₀ is exceptionally substantial. A large drop might indicate a measuring error or a biological anomaly that has to be explained.

Lines 183-184: An explanation is needed if Fm values rise to previously recorded levels above CK.

Paragraph 2.4:

Why does T5 continuously raise potassium, phosphorus, and nitrogen levels?
What causes magnesium to stay constant while other elements change?
Why would the copper and zinc levels increase so much among 60 and 75 days post flowering?

Figure 5 discussion: Why does each treatment reduce calcium accumulation?

Lines 321-322:  “the promotion effect of potassium dihydrogen phosphate (T3-T5) on net photosynthetic 321 rate and transpiration rate decreased with the increase in concentration”  - Contrary to subsequent claims that T4 and T5 enhance photosynthetic performance.

Lines 326-328: Split the text for clarity

Conclusions:

The frequent usage of "significantly" may be an exaggeration in the absence of conclusive statistical support.

The claim that T4 is the most beneficial should be supported by a direct comparison of treatments (T1–T5).
Make it clear that Ci falls while the majority of parameters increase, offering a fair interpretation.

References:

Geographic concentration on China, with less worldwide coverage.

Please, view also the comments in text.

Comments on the Quality of English Language

Dear Editors,

Round 2

Reviewer 1 Report

Comments and Suggestions for Authors

The authors took into account many comments and made significant corrections. I have no further significant comments. Figure formatting and text sizes should be reviewed before publishing.